

# Probabilistic Flood Extent Estimates from Social Media Flood Observations

Tom Brouwer[1,2], Dirk Eilander[1], Arnejan van Loenen[1], Martijn J. Booij[2], Kathelijne M. Wijnberg[2], Jan S. Verkade[1], Jurjen Wagemaker[3]

[1]Deltares, Delft, Boussinesqweg 1, 2629 HV, The Netherlands
[2]Dept. of Water Engineering and Management, University of Twente, Enschede, Drienerlolaan 5, 7522NB, The Netherlands
[3]FloodTags, The Hague, Binckhorstlaan 36, 2511 BE, The Netherlands

*Correspondence to*: Dirk Eilander (Dirk.Eilander@deltares.nl)

**Abstract.** The increasing number and severity of floods, driven by phenomena such as urbanization, deforestation, subsidence and climate change, creates a growing need for accurate and timely flood maps. This research focussed on creating flood maps using user generated content from Twitter. Twitter data has added value over traditional methods such as remote sensing and hydraulic models, since the data is available almost instantly, in contrast to remote sensing and requires less detail than hydraulic models. Deterministic flood maps created using these data showed good performance ($F^{(2)} = 0.69$) for a case study in York (UK). For York the main source of uncertainty in the probabilistic flood maps was found to be the error of the locations derived from the Twitter data. Errors in the elevation data and parameters of the applied algorithm contributed less to flood extent uncertainty. Although the generated probabilistic maps tended to overestimate the actual probability of flooding, they gave a reasonable representation of flood extent uncertainty in the area. This study illustrates that inherently uncertain data from social media can be used to derive information about flooding.

**Keywords:** User generated content, Social media, Twitter, Flood extent estimation, Uncertainty analysis, Probabilistic flood mapping

## 1 Introduction

Between 1995 and 2015 2.3 billion people were affected by floods (UN, 2015), which is about one third of the world's population. Worldwide developments such as urbanization, deforestation, subsidence and climate change are expected to increase the occurrence of floods and number of people affected by them. This creates a growing need for timely and accurate information about the locations and severity of flooding. In multiple phases of the disaster management cycle (Carter, 2008), this information is useful. In the mitigation phase, data about previous flood events can be used to evaluate the probability of flooding and prevent urban expansion into flood prone areas. If flood prone areas are already inhabited, information about flood risk can also be used to improve disaster preparedness. In the response phase, information about the current flood situation is useful, for example for rescue workers who want to identify affected areas and assess the accessibility of roads.



Finally, in the recovery phase, flood information can help insurance companies in evaluating flood damages and aid organizations in targeting rebuilding efforts.

Traditionally, flood information in the form of flood maps has been produced using either hydraulic models or remote sensing. Applying these in real-time however, may be problematic. Hydraulic models require a detailed schematization of the study area, knowledge about the cause of a flood and may require considerable computational time. Also forecasts of input data, such as discharge or precipitation may not be readily available. Remotely sensed data may take several hours to become available (Mason et al., 2012) and its temporal resolution is often limited (Schumann et al., 2009).

Applying user generated content, which is content generated by users of online platforms such as blogs, wikis and social media, has recently gained momentum in flood mapping. Recent studies focussed specifically on using social media content. Platforms such as Twitter, Facebook and Flickr produce large amounts of real-time data that may be used to derive information about flooding. In coarse scale applications for example, these data can be used to detect the occurrence of a natural disaster (Earle et al., 2011). On a more detailed level these data can also be used to assess the extent of the disaster. In the context of flood mapping, some investigations merely used these data as auxiliary data. Examples include the assessment of the accuracy of remote sensing derived flood maps (Sun et al., 2015) and the selection of the most realistic result of a series of hydraulic model runs (Smith et al., 2015). Others actually created flood maps directly from the data. Schnebele et al. (2014) used the density of flood related Twitter messages (Tweets) to get an indication of flood extent and in the PetaJakarta project, the number of Tweets in an area is used to indicate flood severity (Holderness & Turpin, 2015). Fohringer et al. (2015) created flood maps by interpolating water levels which were manually derived from photographs posted on social media. Eilander et al. (2016) on the other hand used an automatic method to derive water depths and locations from Tweets, and created flood maps using a flood fill algorithm.

The aforementioned studies all focussed on obtaining flood extents from social media content. These flood extents, however, did not relay information about uncertainty, even though uncertainty is an inherent characteristic of information derived from social media content. Locational information of Tweets for example can be uncertain, because geo-tags are available for only a very small number of Tweets and may deviate from the actual location of the observation (Hahmann et al., 2014). McClanahan & Gokhale (2015), who derived locations from the text in Tweets, indicate that the locations they derived from messages in New York City had an average error of 1.72 km. Eilander et al. (2016) were the first to give an estimate of the likelihood of areas being flooded, by harvesting Tweets. This likelihood was based on the number of Tweets found for individual administrative areas, rather than knowledge about the actual errors in the data used.

Information about uncertainties can help in assessing the quality of generated flood maps. In addition, it can also serve as an information source of its own. This was for example the case in the search for Air France flight 447, which disappeared over





the Atlantic Ocean in 2009. Probabilistic maps of the location of the wreckage were successfully used to find the wreckage in 2011, while previous attempts, spanning a two-year period, all failed (Stone et al., 2014). More specific to flood mapping, information about uncertainties can be used to direct surveys to areas in which the flood extent is highly uncertain. The information can also be used by rescue workers navigating an affected area, to choose the most optimal route by weighing the

length of a route against the probability of it being flooded.

In the present manuscript we investigate the applicability of using social media content from Twitter to generate probabilistic flood maps. We explicitly address uncertainties in the data and assess the added value of probabilistic maps over deterministic maps. The analyses presented in this manuscript give insight into the magnitude of errors in flood observations derived from

Tweets and improve understanding of how these errors affect the flood extent estimates. Furthermore, we investigate how the uncertainty caused by errors in the Twitter data relates to the uncertainty caused by other sources of error.

This manuscript starts with a description of the case study (Sect. 2). This is followed by an overview of the methods used (Sect. 3). Section 4 subsequently presents the research results and Sect. 5 includes an in depth discussion thereof. Finally, the

conclusions of the research are given in Sect. 6.

## 2 Case study

On the 27th of December 2015 peak water levels on the river Ouse, caused by large amounts of rainfall over the month December, led to the flooding of a considerable area within the City of York in the North of England. The 2015 floods were widespread and inundated almost 600 homes and businesses (Stott, 2016).

The flooding of the City of York in December 2015 was selected as a case study, since both high resolution terrain elevation data, as well as recorded flood extents were available for this event. In Fig. 1 an Environment Agency (EA) digital terrain model (DTM) of the city of York is given (EA, 2014). This manuscript focusses on Central York, delineated by the central administrative areas of York. In most of the study area terrain slopes are moderate, although some higher ridges are found in

the south and south-west of the area. North of these ridges the inner-city of York is located. At this location there is a confluence of several rivers, of which the River Ouse is the largest.

## 3 Methods

Data from Twitter were used to derive flood information, since these data are openly and freely available. The research was comprised of several phases. First we extracted useful information from flood-related Tweets. This information was

subsequently used to create a deterministic flood extent estimate. After this, based on information about the magnitude of



errors in the data, probabilistic flood extent estimates were derived. In this section we discuss the setup of these three phases along with the methods used to evaluate the accuracy of the deterministic and probabilistic flood maps.

## 3.1 Twitter data extraction

We performed several steps to create a database of Twitter based flood observations (Fig. 2). First of all, we collected all
Tweets that contained a number of common flood-related keywords such as 'flood' or 'inundation', using the Twitter streaming API. From this database we selected the Tweets that were sent between the 25th and 30th of December 2015. To ensure purely Tweets regarding York were found, only Tweets that mentioned 'York' or '#YorkFloods' were included and messages referring to 'New York' or 'York County' (both in the USA) were excluded. As a last selection step, we only kept Tweets that contained explicit references to locations, such as streets or points of interest (POIs), by looking for common keywords such
as 'street', 'lane', 'museum' or 'school'. Some other minor filters were applied to ensure only relevant Tweets were found, for example by excluding Tweets related to flood barriers and flood warnings.

We derived locations from the Tweets in the remaining dataset by manually identifying the section of the Tweet that contained a locational reference. Based on this reference, X and Y coordinates were assigned to the Tweets. To illustrate this process,
the following Tweet is used as an example:

*"Cumberland Street in York - say they're used to flooding here but only 2000 was worse" (By: @jimtaylor1984)*

The locational reference in this Tweet is *"Cumberland Street"*. These locational references were used directly to search Google
Maps. The message above however did not refer to a point location, which would be the case for a POI, but to a line element. We derived exact spatial coordinates from such Tweets by using the location of the street from Google Maps in combination with the DTM of the EA (EA, 2014). If topographical depressions were present along the street, the deepest depression, identified by filling the sinks in the DTM, was used as the location of the observation. In case no depressions were found, the point of lowest elevation was used, which was the case for the Tweet above (See Fig. 3).

To review the accuracy of the spatial coordinates derived from the Tweets, we looked at the photographs attached to some of the Tweets and compared them to Google StreetView. If we found the specific location of a photograph, we compared it to the spatial coordinates derived from the Tweets text to determine the locational error (example: Fig. 3).

## 3.2 Flood extent mapping

Flood extents were derived at 20 m resolution using the locations derived from the Twitter messages and a DTM from the EA (EA, 2014). We applied an interpolation method to derive flood maps from the observations, similarly to Fohringer et al. (2015). Before interpolation, two processing steps were applied to guarantee more realistic flood extent maps. Firstly, we





derived water levels relative to the nearest drainage channel from each observation. Since none of the Tweets about York mentioned water depths, water levels were derived by assuming the same default water depth (DWD) for all observations. We calculated these water levels relative to the nearest drainage channel by using a height above nearest drainage (HAND) elevation model (Rennó et al., 2008; Norbre et al., 2011). Secondly, observations were grouped based on the local drainage

directions (LDDs) to interpolate only hydrologically 'connected' observations. Instead of using the bilinear spline interpolation of Fohringer et al. (2015), we used inverse distance weighting (IDW) interpolation to determine the flood extent. Figure 4 gives an overview of this process. These steps are further explained in this section.

Norbre et al. (2016) applied the HAND concept to derive inundation extents for fluvial floods. By using a HAND map instead

of a DTM, river slopes are filtered from the dataset. This means that HAND values in an area are directly related to river stage and HAND contour lines describe the flood extent at a specific river stage. Since river slopes are filtered, upstream observations are also less likely to cause overestimations of water levels downstream. We constructed a HAND map from the DTM by deriving the LDDs and using these to determine the elevation value of each grid cell in the study area, relative to the nearest drainage channel. Grid cells were identified as being on a drainage channel if they had an upstream area of 6 km$^2$ or more,

which gave the best representation of drainage channels in the study area. Topographical depressions were filtered to derive the LDDs used to construct the HAND map. To also account for pluvial flooding of local topographical depressions, these depressions were reintroduced in the final HAND map. This map was then used to translate the DWD assigned to each observation to a water level with respect to the nearest drainage channel.

In order to only interpolate observations belonging to the same continuously flooded area, we grouped observations prior to interpolating the water levels. This was done by calculating the LDDs in the area using the DTM. These LDDs were used to determine which cells are downstream of an observation. If the location of the observation is flooded, it is assumed its downstream cells are also flooded, since these are located lower than the observation and are directly connected to it. Therefore, all observations of which the flow paths intersect downstream, are located within the same continuously flooded area.


The water levels (relative to the nearest drainage channel) of each group of observations were subsequently interpolated using IDW interpolation as given by Eq. (1) and Eq. (2):

$$Z_{x,y} = \frac{\sum_{i=1}^{n} z_i * W_i}{\sum_{i=1}^{n} W_i} , \tag{1}$$


$$W_i = \frac{1}{(d_{x,y,i}+s)^p} , \tag{2}$$





Where $Z_{x,y}$ (m) is the interpolated water level at spatial coordinates x and y, $Z_i$ (m) is the observed water level of observation $i$, n is the total number of observations, $W_i$ is the interpolation weight of observation $i$, $d_{x,y,i}$ (m) is the distance to observation $i$ measured along the flow paths downstream of observations, s (m) is the smoothing parameter and p is the power parameter.

We used IDW interpolation since it allows for smoothing, which is useful in averaging the errors in social media content. Also the nominator and denominator of Eq. (1) can be updated with new observations, meaning the additional computational time in real-time applications is limited. Water levels of observations were interpolated along their downstream flow paths. The water levels along these flow paths were subsequently given to the grid cells upstream of these flow paths, which produced a grid of water levels. From this we subtracted the HAND map, to create a grids of water depths in the area. Since water depths

were calculated in areas which were separated from the observations by small ridges, flooded areas that were not connected to any of the observations were removed. This procedure produced the deterministic flood maps.

### 3.3 Uncertainty analysis

The uncertainties in the flood extent maps were investigated using a Monte Carlo analysis. We evaluated the uncertainty originating from errors in the locations derived from Tweets, errors in the elevation data, uncertainty in the parameters of the

IDW equation and uncertainty in the DWD. The characteristics of the error distributions used to simulate these errors are given in Table 1.

The analysis of the locational errors of Tweets indicated that the locations derived from Tweets that refer to point locations contain less error than those derived from Tweets that refer to streets (see Sect. 4.1). The locational errors of both types of

Tweets were therefore simulated differently. We simulated the locational errors of Tweets that referred to point locations by adding random errors to the spatial coordinates of the Tweets. The locational errors of Tweets that referred to streets were simulated along these streets. We did this by extracting the streets to which the Tweets referred from OpenStreetMap. The locational error was modelled using a normal distribution. To generate each realization, the observations were moved a distance along the street, which was drawn from this distribution. As some streets are shorter than six standard deviations, effectively

reducing the modelled error, the standard deviation used for modelling these errors was modified so that the resulting errors matched the observed locational errors.

Since no reference information was available regarding the accuracy and errors in the EA DTM, these errors were simulated using typical values from literature. Since using independent normally distributed errors does not accurately reflect errors in

the elevation data (Heuvelink et al., 2007; Raaflaub & Collins, 2006), spatially auto correlated errors were added using the method described by Dullof & Doucette (2014). Based on typical values of standard deviations and auto correlation distances of errors in LIDAR elevation data found in literature (Leon et al., 2014; Mudron et al., 2013; Li et al., 2011; Livne & Svoray, 2011; Hodgson & Bresnahan, 2004) a standard deviation of 20 cm and correlation distance of 100 m was used. These errors



were added to the original 2 m resolution DTM, before resampling it to 20 m resolution and creating the corresponding HAND map.

The uncertainty caused by the input parameters, being the power and smoothing parameters (Eq. 2) and the DWD, was also

evaluated. Since no specific information is available about the distribution of errors in these parameters, we used a uniform distribution. The power parameter was varied between 2 and 5, the smoothing between 0 and 2000 m and the DWD between 20 and 80 cm.

To determine the number of Monte Carlo simulations required to produce the probabilistic flood maps, multiple maps were

created using the same input uncertainties. It was found that using 1000 Monte Carlo simulations, two probabilistic flood maps generated using the same input error distributions were nearly identical.

### 3.4 Evaluation of results

We evaluated the accuracy of both the deterministic and probabilistic flood maps using recorded flood extents. A draft version of the fluvial flood extents for the City of York was supplied by the EA. These flood extents however only identified areas

that were directly affected by flooding from the rivers. Areas separated from the river around Knavemire Road, Water Lane and Shipton Road were also known to be flooded based on photographical evidence in some Tweets. This was confirmed by news articles about these three locations. The flood extents around these locations were approximated by using the EA dataset of historic flood extents (EA, 2015) from between the years 1991 – 2012. The flood extent specific to fluvial flooding during the December 2015 event and the historical flood extents were merged into one flood extent map for validation. Using these

data, we evaluated the accuracy of the deterministic flood maps by calculating the $F^{(2)}$ statistic of Aronica et al. (2002). This statistic gives the percentage of correctly modelled flooded area, relative to the total area that is either in the modelled or observed flood extents.

We evaluated the probabilistic flood maps using reliability diagrams. These diagrams offer a comparison between the modelled

probability (on the horizontal axis) and the observed probability (vertical axis) of flooding (Wilks, 2006). If the probabilistic flood map is correct, it is expected that of all cells on the map that have for example a 10 % probability of being flooded, only 1 out of 10 cells is actually flooded in reality. To create the reliability diagrams, the modelled probabilities were first binned in 10% intervals. For each probability bin, the cells on the probabilistic map that fell within that bin range were compared to the map with actual flood extents (the validation map). The number of times the selected cells were indeed flooded in reality

were counted and divided by the total number of cells in this bin. For each binned probability interval, the central value was taken as the modelled probability value that should be compared to the observed probability. Note that the first probability bin ranged from 0.01% to 10%, and the 0% probability of flooding value in the diagram included all cells having less than 0.01% probability of flooding. In case the line that is constructed this way is exactly on the diagonal of the graph, the probabilistic





flood map gives an accurate representation of the actual probability of flooding. In case the line is above the diagonal, flood probabilities are underestimated and in case it is under, flood probabilities are overestimated.

We assessed the relative importance of the different sources of error on the uncertainties in flood extent by creating three
different uncertainty estimates: one by only simulating locational error, one by only simulating errors in the DTM and one by only simulating errors in the parameters. For every uncertainty estimate, the $F^{(2)}$ statistic of each random simulation was calculated. We used these values to derive three empirical cumulative distributions of the $F^{(2)}$ statistic for the uncertainty estimates generated by simulating the individual sources of error. These were used to review the relative importance of the different types of errors for the accuracy of the maps.

**4 Results**

During the York floods, 8,000 unique flood related Tweets, posted between the 25th and 30th of December 2015, were harvested. Using the process discussed in Sect. 3.1 a database of 160 Tweets was constructed. Only from 87 of these could a location be derived from the text of the message. Although 56 of these Tweets had photographs attached, we could only match 26 of them to a location on Google StreetView. These were used to assess the quality of the locational references derived from
the text of Tweets.

**4.1 Locational errors**

We compared the locations that were derived from the text in the Tweets and used to create the flood maps to locations derived from attached photographs, in order to evaluate the magnitude of locational errors. Figure 5a gives the result of this comparison. The magnitude of locational errors depends heavily on the type of locational references in the Tweets. If point locations such
as POIs or intersections are mentioned in the Tweets, the locational error is limited (Fig. 5c). If streets are mentioned in the Tweets however, locational errors are considerably larger (Fig. 5b). The difference is likely caused by the fact that the locations of point references were directly extracted from Google Maps, whereas an additional procedure was necessary to derive exact spatial coordinates from Tweets referring to streets. The large outlier in Fig. 5b is for example a Tweet that refers to Huntington Road, which is a long street that is located alongside the River Foss (see Fig. 1). The photograph was made at the northern end
of Huntington Road, whereas the Tweet was pinpointed to the deepest depression along the road, in the south. There were however photographs attached to other Tweets which indicated that this southern location was also flooded. Without this outlier, the standard deviation in locational errors of Tweets referring to streets reduced to 118 m.





## 4.2 Flood Extent mapping

We created deterministic flood extent estimates by interpolating the locations and water levels derived from the Tweets. The parameters of the IDW interpolation were calibrated using the $F^{(2)}$ value based on the EA recorded flood extents. A power parameter of 4 in combination with a smoothing of 600 m (Eq. 2) and a DWD of 50 cm gave the best results. Figure 6 gives a comparison between the flood extent generated using information harvested from the Tweets and the validation data, by classifying all grid cells based on the four quadrants of a confusion matrix. An $F^{(2)}$ value of 0.69 was found, indicating that the modelled area of the flood extent that is correct, makes up 69% of the total flood extent either in the modelled or observed data.

The flood extent estimate is correct for a large part of the inner-city (location [1]). Even at smaller flooded areas, such as the ones north-west and south-east of location [2] a good estimate of flood extent is generated. The added value of separating groups of observations that are not in the same flooded area is seen at area [3]. Without separating observations, the underestimation of flood extent in this area would be considerable, whereas separating the observations results in a much better flood extent estimate for this area. Although at some locations minor underestimations of flood extent are seen, there is only one large area missing at location [4]. However, no observations of flooding close to, or in this area were found in the Twitter dataset. This underestimation is therefore a result of the lack of data rather than an error of the interpolation method.

## 4.3 Uncertainty Analysis

We created probabilistic flood extent estimates by varying the input parameters as well as simulating the locational errors and errors in the DTM in a Monte Carlo analysis. Based on the results in Sect. 4.1 the error distance along streets was modelled using a normal distribution with a standard deviation of 200 m. This modelled error effectively translates into a standard deviation in spatial coordinates of 100 m as some streets were too short to reproduce the full error distribution. Given the results from Fig. 5c errors in point locations were simulated using a normal distribution with a standard deviation of 50 m. The uncertainty resulting from simulating these locational errors along with errors in the DTM and parameters is given in figure 7a.

The uncertainty in flood extent is considerable (i.e.: the flood probability is around 50%). However, near the inner-city, at location [1], the uncertainty is limited. This is only partly caused by the high density of observations in this area and is mostly a result of the fact that the inner-city of York is situated lower than its surroundings, effectively limiting flood extents. For the areas within York that are more flat, the uncertainty in flood extent was generally larger. The density of observations is not well represented in the uncertainty estimates. Generally speaking, one would expect an area that has a high probability of flooding to have multiple observations in it, since a single observation can be placed there due to the Tweet being misinterpreted. At location [3] however, there is a large area with a high probability of flooding, even though only one





observation is pinpointed to it. Location [3], as well as location [2] had high probabilities of flooding, although they were not flooded in reality.

We assessed the performance of the probabilistic flood extent map in Fig. 7a by comparing it to the validation data and constructing a reliability diagram (Fig. 8a). Probabilities between roughly 15 to 85% are mostly overestimated, although the most important probabilities close to 0 and 100% are accurately represented in the map. Comparing this map of all uncertainties to a map created by simulating only the errors in the elevation data and the parameters (Fig. 7b) indicates that locational errors are likely responsible for a considerable amount of uncertainty in the flood extent estimates. This is further confirmed by the results in Fig. 9, which shows the empirical cumulative distribution functions of the $F^{(2)}$ measure of accuracy. This was calculated by using the result of each random simulation of the Monte Carlo Analyses of different types of errors separately. It can be clearly seen that locational errors cause most variation in the accuracy of the maps.

However, the reliability diagram that was constructed using the map generated without simulating errors in location (Fig. 8b), shows that by omitting the simulation of locational error, the uncertainty calculated using the Monte Carlo analysis more accurately describes the real uncertainty in flood extent. This indicates that either the probability distributions used to simulate these errors or the way these errors are propagated cause the flood probability to be overestimated.

## 5 Discussion

This study gives insight into the potential of using inherently uncertain social media content to create flood maps. The deterministic flood map created using social media content already gave a reliable estimate of flood extent. However, the magnitude of locational errors was considerable, and the analyses presented in this manuscript indicate that these locational errors cause considerable uncertainty in flood extent.

This uncertainty in flood extent seems to be strongly related to the topography of the study area. Especially at locations with moderate terrain slopes, flood extent uncertainty was limited. Flat areas tend to have a larger uncertainty in flood extent, since the differences in water levels caused by the different sources of error, cause larger horizontal changes in flood extent at these locations.

By simulating all sources of errors however, the uncertainty in flood extent was overestimated, which can have a variety of reasons. Firstly, the magnitude of locational errors may have been overestimated by calculating the errors based on attached photographs. For example, photographs can be taken at a different location, because the location that is mentioned in the text is too severely flooded. Also in places where locations of flooding were omitted, locational errors were overestimated. The outlier, mentioned in Sect. 4.1, illustrates this. This Tweet referred to a street being flooded. The location that was derived





from the Tweet, was known to have flooded based on photographs attached to other Tweets. Nevertheless, a large locational error was calculated, because the photograph attached to the Tweet showed that a second location along the street had also flooded. The procedure used to derive locations from Tweets referring to streets was only able to identify one location along the street, causing the second location of flooding to be omitted. Since this is an error of omission, rather than a locational

error, the exclusion of the outlier is believed to have given a better estimate of the standard deviation in locational errors.

Another cause of the overestimation of uncertainty might be found in the empirical probability distribution used to simulate locational errors. The normal distribution used does not reflect the sharp peaks seen at 0m in the graphs of Fig. 6. Also, using a conventional error distribution to simulate errors might not give a correct representation of the actual errors in location. In

reality it is more likely that an observation originated from a lower location or a topographical depression, whereas purely using random errors can place observations on top of hills, which are unlikely to be flooded.

We expect however that the main reason for the overestimation of total uncertainty is caused by an overestimation of the errors in either the DTM or parameters. This can both be a result of the quantification of these sources of errors or the methods used

to assess their impact. It is likely the quantification of parameter errors contributed most to this overestimation of uncertainty, since these were quantified conservatively in absence of accurate information about their error distribution.

In addition to improving on the quantification and simulation of the different types of errors, the uncertainty maps can be further improved by including information about the density of observations. By default, the errors of all observations are

drawn from the same error distribution in the Monte Carlo simulation, although observations that belong to large clusters are likely more certain than observations that are completely isolated. That the density of observations was not accurately represented caused large areas to have a high probability of flooding, although they contained only very few observations.

The probabilistic maps proved to be a useful addition to the deterministic map. Firstly, the uncertainty estimates are a source

of information in itself. For example, where the deterministic map contained an underestimation of flooded area at location [4] (Fig. 5), the uncertainty estimate showed that flooding was highly uncertain at this location. This information can be used to send personnel into the area, to find out if the area is really flooded and thereby reduce the uncertainty at that location. Similarly, the probabilistic map confirmed the accuracy of the deterministic map near the inner city of York. Furthermore, the probabilistic maps provide information about the flood extent without the need for prior calibration of the model parameters.

Therefore, these maps can potentially provide real-time flood extent information, without having to calibrate the method to that particular event or location first. However, to understand how the modelled uncertainty relates to the observed flood extent for a particular area or event, some validation might still be required.



Further research is necessary to explore the full potential of the uncertainty estimates of flood extents derived from social media. First of all, the consistency of the results for other events or locations should be further reviewed. By reviewing more case studies, the effect of topography on flood extent uncertainty and the (in)dependency of the model parameters for a specific event can also be further investigated.

Such additional studies can also guide further optimization of the uncertainty estimates. If the results of further case studies indicate that the probability of flooding is consistently overestimated by the same amount, post-processing of the maps, by using the information from the reliability diagram, can be an easy way to improve the uncertainty estimates. If this is not the case, improving the maps by improving the process of simulating locational uncertainties might lead to better results. The

overestimation of uncertainty caused by errors in the elevation data and parameters should then also be further reviewed. One element that must be improved, regardless of the results of executing multiple case studies, is the inclusion of the density of observations in the uncertainty estimates. Also methods to include important barriers in the area in the coarse resolution DTM should be investigated, since it was observed that some areas were erroneously assigned a high probability of flooding. Although such improvements can also be achieved by using actual higher resolution data, this will seriously affect

computational time and thereby the potential of real-time application of the maps.

Where current methods for flood extent mapping such as hydraulic models and remote sensing have shortcomings in real-time application, this is where the real potential of using social media content lies. The methods used in this report can potentially be applied in real-time. Random simulations for the York case were generated at a pace of about 100 simulations per minute,

and the fact that calculations for single observations can just be added to the nominator and denominator of Eq. 1 makes that time can be gained if only the effect of an additional observation has to be added. To further reduce computational time, also the use of different sampling techniques, that can potentially reduce the number of random simulations necessary, can be reviewed. Besides computational time, a further look into the gathering of observations is required for a real-time implementation. Especially in real-time application, a high number of observations is necessary to ensure an up to date map

can be made at any point in time. Using the selection techniques applied in this paper, only a small number of Tweets was found. It should be reviewed if using different search techniques or additional sources of data can improve the number of observations, or if techniques such as crowd interaction have more potential in increasing the number of observations available for creating the maps.

## 6 Conclusions

This study illustrates that social media content has real potential in generating flood extent estimates. Although errors in locations derived from the Tweets were considerable, the deterministic flood extent map presented in this manuscript showed



good agreement with validation data. The deterministic flood maps therefore can be used to gain insight into the current situation of flooding.

Using information about the errors in the Tweets, DTM and parameter settings, we managed to construct a probabilistic flood extent map. The uncertainty in flood extent mainly originated from the locational errors of Tweets, whereas DTM and parameter errors contributed less to flood extent uncertainty. A comparison of the probabilistic map to validation data showed that by simulating errors in the Tweets, DTM and parameters, a reasonable estimate of flood extent uncertainty is generated, which provides users with additional information on top of the deterministic flood map.

These results illustrate that social media content can be used to derive information about floods, regardless of the uncertainties in this content. If further improvements are made, so that the methods used in this report can be applied in real-time, these maps have the potential of filling in the gap where hydraulic models and remote sensing are lacking.

**Code Availability**

The analyses in this manuscript were performed using Python 2 scripts. The code used for the different analyses is publically available on GitHub and published in the Zenodo research data repository (doi: 10.5281/zenodo.165818).

**Data Availability**

In the study data from downloaded from the Twitter API as well as data from the Environment Agency was used. The filtered subset of Tweets used in the research, the information about streets extracted from OpenStreetMap as well as the 20m resolution DTM and HAND map can be found in the aforementioned GitHub project (doi: 10.5281/zenodo.165818). Also the data used to create the plots and maps are available at this location.

**Competing Interests**

The authors declare that they have no conflict of interest.

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

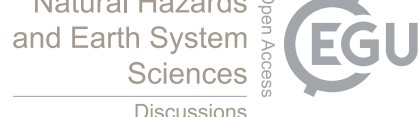



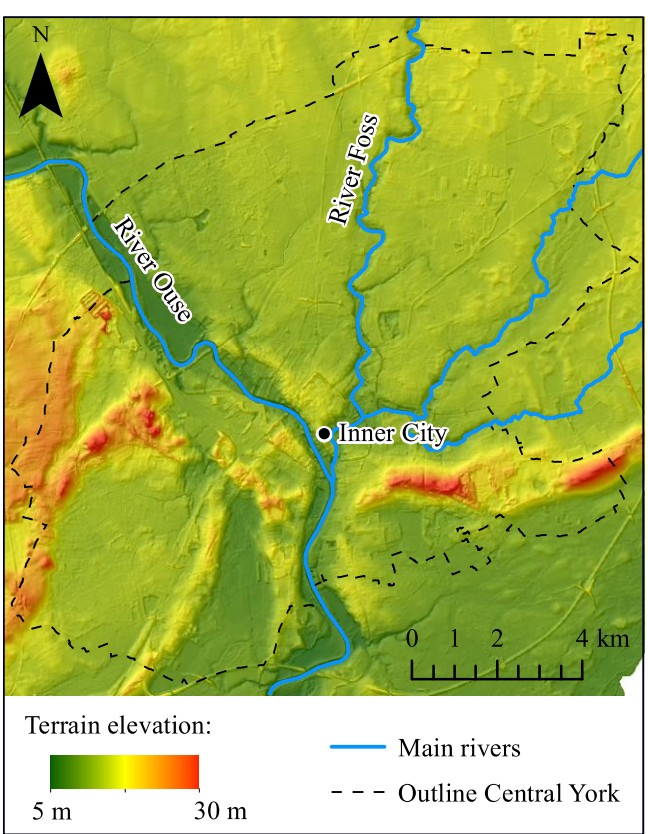

**Figure 1.** Digital Terrain model of the York study area

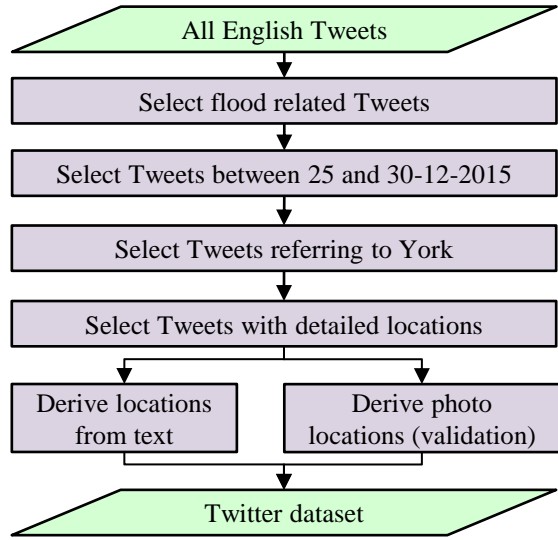

5   **Figure 2.** Process of constructing the dataset of Tweets




*"Cumberland Street in York - say they're used to flooding here but only 2000 was worse"*

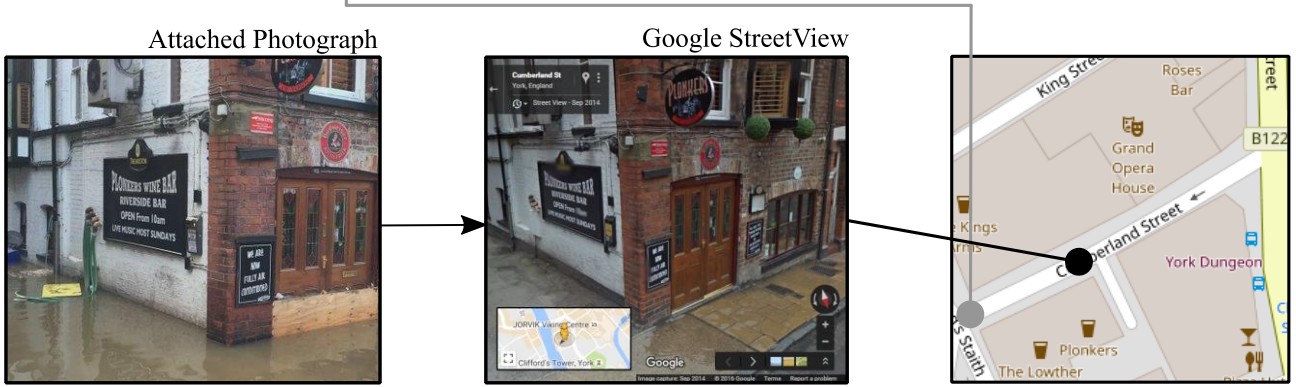

Attached Photograph            Google StreetView

**Figure 3.** Example of determining the error in the spatial coordinates derived from the text of a Tweet, based on an attached photograph. The grey dot is the location derived from the text of the Tweet, and the black dot is the location derived from the attached photograph.

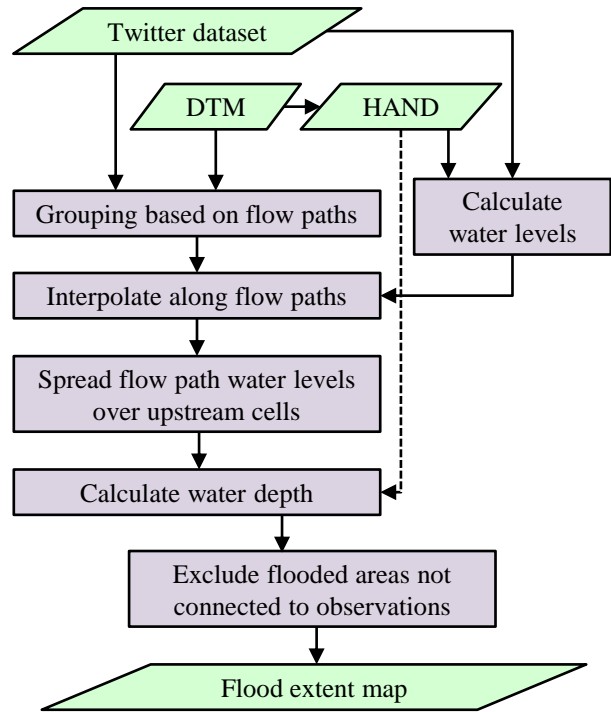

5    **Figure 4.** Process of creating flood extent maps



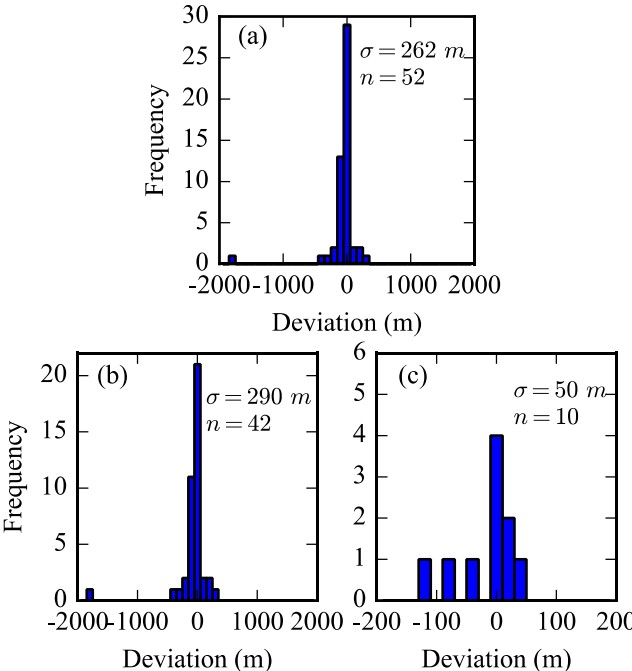

**Figure 5.** Locational errors in X/Y coordinates of all Tweets (a), only the ones referring to streets (b) and only the ones referring to point locations (c)





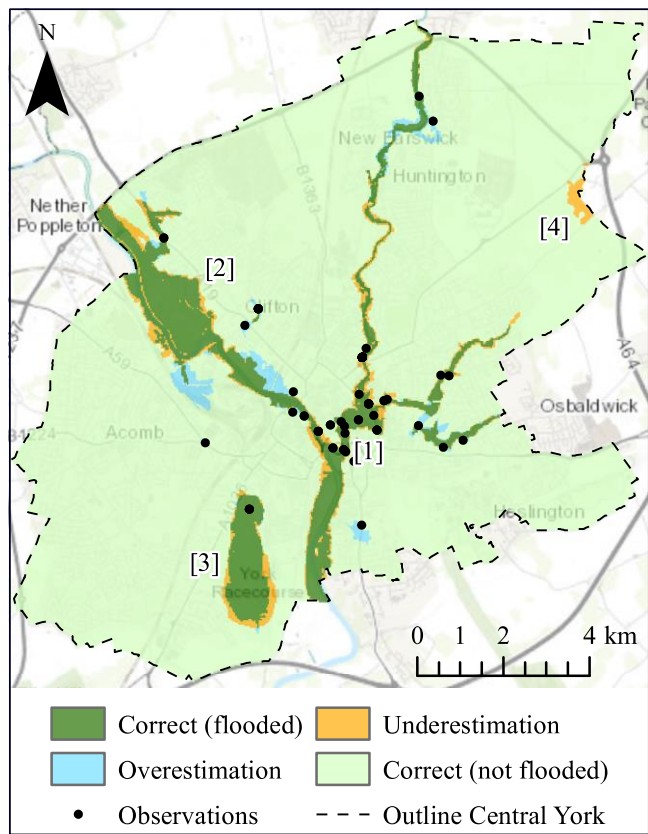

**Figure 6.** Deterministic flood map classified as the four quadrants in a confusion matrix. Locations [1] to [4] are discussed in text.




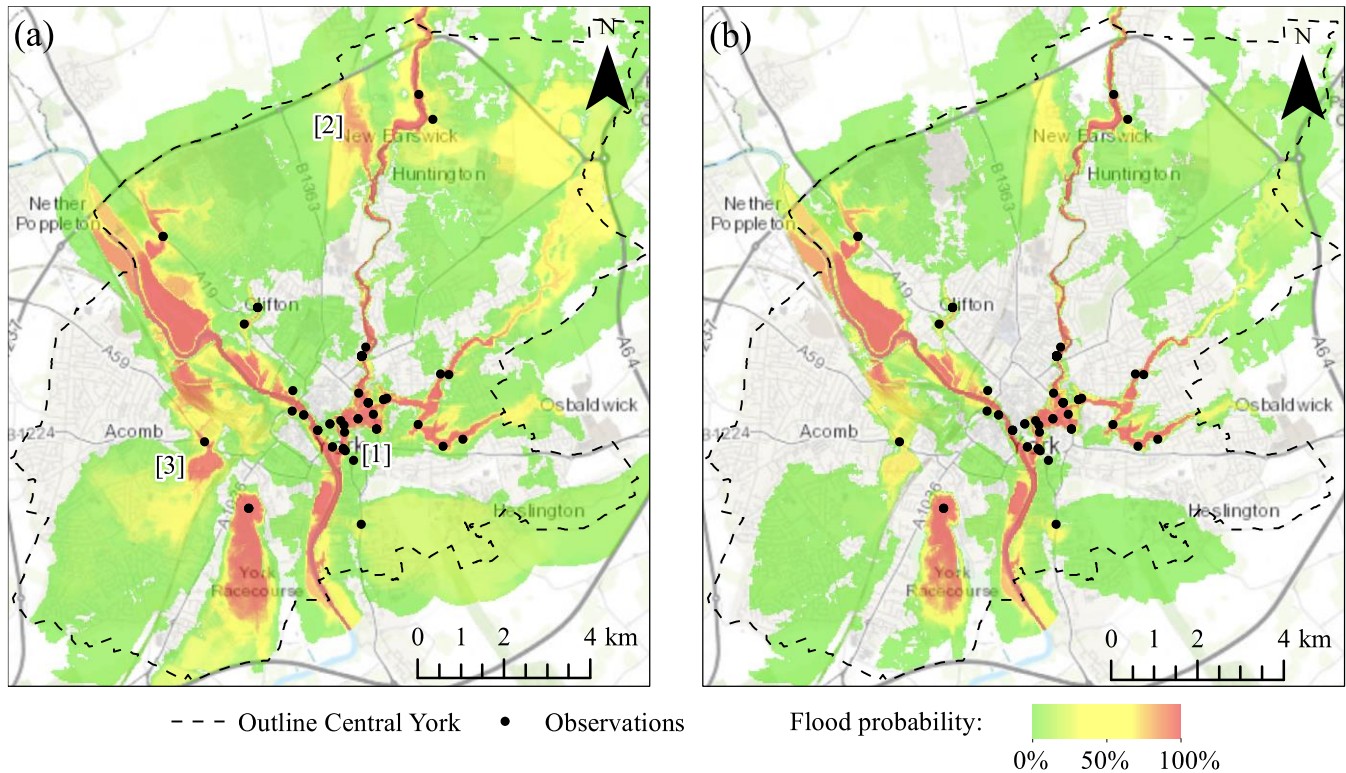

**Figure 7.** Probabilistic flood map generated by simulating locational errors, errors in elevation data and errors in parameters (a) and a probabilistic map generated simulating errors in the elevation data and parameters only (b).

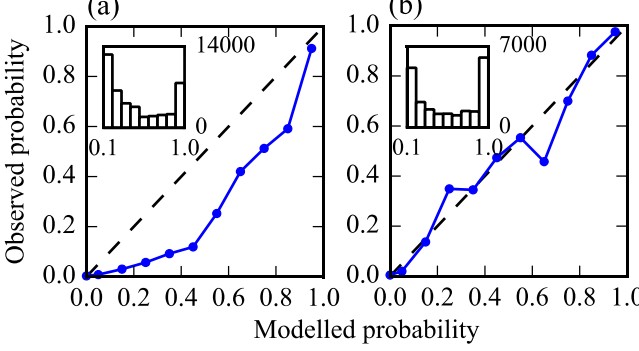

**Figure 8.** Reliability diagrams constructed from the probabilistic flood map generated by simulating all errors (a) and by simulating only errors in the elevation data and parameters (b). The small histograms give the number of cells within each 10% bin of modelled flood probability.



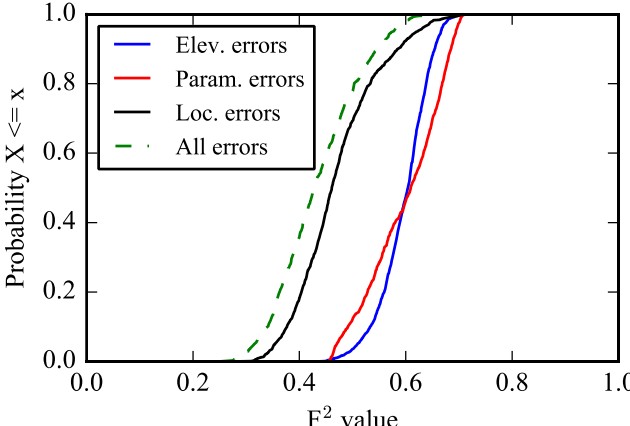

**Figure 9.** Empirical cumulative distribution functions of the $F^{(2)}$ performance statistics derived by simulating purely errors in the elevation data (blue), errors in the parameters (red), locational errors (black) and the combination of these errors (dashed green).

5    **Table 1.** Quantification of error sources. [a]See Sect. 4.2

| Error source | Distribution | Parameter | Value |
|---|---|---|---|
| Elevation data | Normal (spatially auto correlated) | μ: | 0 m |
|  |  | σ: | 0.2 m |
|  |  | Corr. distance: | 100 m |
| Tweets (point location)[a] | Normal (X/Y coordinate) | μ: | 0 m |
|  |  | σ: | 50 m |
| Tweets (Street location)[a] | Normal (along street) | μ: | 0 m |
|  |  | σ: | 200 m |
| Power parameter | Uniform (integers only) | Lower bound: | 2 |
|  |  | Upper bound: | 5 |
| Smoothing parameter | Uniform | Lower bound: | 0 m |
|  |  | Upper bound: | 2000 m |
| DWD | Uniform | Lower bound: | 0.2 m |
|  |  | Upper bound: | 0.8 m |