# Peer review of "Probabilistic Flood Extent Estimates from Social Media Flood Observations"

_Natural Hazards and Earth System Sciences, 2016_

## Referee Comment (RC1) · Anonymous Referee #1 · 26 Dec 2016

This article presents an update methodology to generate probabilistic flood maps from social media content, addressing the uncertainty in a detailed way. The topic is of sure interest to NHESS, and the article is quite well structured, but it needs some reworking, for which I suggest a moderate revision.

I suggest an authors' revision of the introduction and the presentation of the case study section since some parts are few explained and some sentences are difficult to read.

Introduction:

page 2; line 10-23. The authors said that social media content gained much attention in flood mapping. For this reason it could be useful to add a sentence on the use of Facebook and Flickr (with references) in this context since only Twitter based studies are addressed.

[Figure]

page 2; line 10-11: please rephrase.

page 2; line11: "focus" at the past participle can take either double or single s, with the single option being highly preferred. Please consider changing it throughout the text.

page 2; line14: The authors could be more precise when citing that "these data can be used to assess the extent of the disasters"? Can you provide some citation? In addition it would be useful to differentiate extent of the area, extent of damage and extent of human losses.

page 2, line 15: I would delete the world "merely" since it seems give a negative connotation to the sentence.

page 2; line 25: relay or rely? Please revise it carefully

Case study:

page 3; line 17: can you be more precise on the amount of rainfall?

page 3; line 17: over the month "of" December. Please consider adding the preposition "of".

page 3; line 18-19: "The 2015 floods" mentioned by Scott (2016) were registered all over UK or within the study area? Please consider rearranging this part by adding more information of the flood event considered (27th of December), adding some information about the damages and the area involved.

page 3; line 25: "North of these ridges..", please rephrase this sentence

According to the methodology some issues need to be fixed, but first of all a general rearranging of the information in the sub-chapters needs to be done. Some details of the data are not concentrated together but are distributed on several chapters that make the reading a difficult process. In addition, it is not clear how many tweets have been considered and/or point of interest have been derived. Secondly, it misses to mention the vertical accuracy of the DTM and I think that the resampling method from

2 m to 20 m resolution should be stated before (please explain why). Moreover, when the observations have been grouped based on the LDDs have been based on the DEM or have been calculated by flow directions or by POI (point of interest) connected with flow directions? In addition the authors need to justify the use of the IDW technique to determine the flood extent at this stage. In addition try to explain the HAND map with few sentences for non-expert users. Some minor changes are listed as follows.

page 3; line 29: Which "useful information" have been extracted to create a deterministic flood extent estimate?

page 4; line 4: It is correct "to perform a step"? Please consider revising this sentence

page 5; line 20-21: Please rephrase the sentence

The chapter 3.4 "Evaluation of results" needs to be better contextualized and explained. The information inside this part seems to include limitation, problems, methodologies and expected results. I suggest rearranging this chapter.

In addition try to better explain the F(2) statistic expressed at page 7; line 20.

The discussion, should provide some interpretation of the results emerged without wasting entire sentences restating the results (ex, page 12; line 11-12 among others). Please, try to verify it and do not repeat same concepts. In addition, you might relate your work to the findings of other studies by finding crucial information in someone else's study that helps you interpret your own data, or perhaps you will be able to reinterpret others' findings in light of yours. In either case you should discuss reasons for similarities and differences between yours and others' findings.

In addition, please rephrase page 12; lines 2-4.

Please, cite in the text the authors of the script just mentioned with a doi at page 13; line 15.

Some figures need minor revision and a detail description is given as follows:

Figure 1 needs more details of where the study area is located. A smaller map of UK where the study area is highlighted is highly suggested to help readers in locating the area. Terrain elevation needs "m (a. s. l.)".

Figure 5. Please add the unit of frequency.

Figure 6. What's the confusion matrix? Please add in the caption what represent the numbers in square brackets.

Figure 7. Please add in the caption what represent the numbers in square brackets.

Figure 8. I suggest enlarging this figure to be able to appreciate the details.

---

## Referee Comment (RC2) · Anonymous Referee #2 · 19 Jan 2017

This paper introduces a rigorous method for producing deterministic and probabilistic flood maps by assimilating user-created contents harvested in social media (Twitter). This approach is very promising for real-time assessment of flood extent, compared to what remote sensing and hydrodynamic modelling can and cannot do.

This is not the first study of this kind of course, but the method proposed presents some original aspects (especially the consistent interpolation of observations over hydraulically connected areas), and the uncertainty analysis through Monte Carlo simulation of identified and quantified error sources is very interesting.

The method is fully illustrated by application to the December 2015 flood in York, UK. This application case is well presented and explained and it is typical of many other potential applications of the methodology. The potential and limitations of the method

are discussed honestly.

I enjoyed reading the paper which is nicely written, very well structured with high quality figures. The code and data are made publicly available, which is commendable.

Therefore, I have only a few minor concerns, or points of discussion:

1/ What is the F(2) statistic is explained in the text (section 3.4), but not in the abstract. Please add a definition when mentioning F(2) in the abstract.

2/ Explain why the DWD has been estimated to range between 20 cm and 80 cm (section 3.3)

3/ I understand that the (potentially large) uncertainties of the validation data were ignored when building the reliability diagrams. I therefore feel that the uncertainties of the probabilistic maps may be even more overestimated than claimed in the paper. Please discuss this issue, the possible consequences of neglecting the validation data uncertainties, and how they could be included in the analysis.

---

## Author Comment (AC1) · 3 Mar 2017

The authors would like to thank the anonymous referee for his or her review of our article. It is clear the referee has read the article thoroughly, which has led to useful feedback. We are convinced that the changes resulting from this feedback will greatly improve the quality of the article. Each of the comments of the anonymous referee, along with our reaction, and the changes to the article we propose, is discussed in the supplement "Paper_SocialMediaFloodExtentEstimates_ResponseRC1.pdf".

The changes discussed are also included in a 'track-changes' version of the original manuscript. This document is included as a supplement and also contains changes made to the article in response to the feedback of the second reviewer as well as some additional minor changes.

[Figure]

Please also note the supplement to this comment:
http://www.nat-hazards-earth-syst-sci-discuss.net/nhess-2016-376/nhess-2016-376-AC1-supplement.zip
* * *

---

## Author Comment (AC2) · 3 Mar 2017

The authors would like to thank the anonymous referee for his or her enthusiastic response to our article. We are convinced that the proposed changes will greatly improve the paper. Our responses to the comments of the anonymous reviewer, along with the changes to the article we propose, are discussed in the supplement "Paper_SocialMediaFloodExtentEstimates_ResponseRC2.pdf".

The changes discussed are also included in a 'track-changes' version of the original manuscript. This document is included as a supplement and also contains changes made to the article in response to the feedback of the first reviewer as well as some additional minor improvements.

Please also note the supplement to this comment:
http://www.nat-hazards-earth-syst-sci-discuss.net/nhess-2016-376/nhess-2016-376-AC2-supplement.zip

---

## Author Response (AR1)

**Response to author comments and list of other changes made in "Probabilistic Flood Extent Estimates from Social Media Flood Observations" by Tom Brouwer et al."**

The authors would like to thank the anonymous referees for their reviews of our article. It is clear both referees have read the article thoroughly, which has led to useful feedback. We are convinced that the changes we made based on their feedback, have greatly improved the quality of the article.

This document contains a list of our responses to the comments of the first (RC1) and second (RC2) anonymous referees.

10  Our responses to the referee comments and the changes we made based on these comments, as well as some additional improvements we made are listed below.

All changes to the article discussed below, can be found in the 'track-changes' version of the document, which can be found at the end of this pdf.

**Response to RC1**

*"page 2; line 10-23. The authors said that social media content gained much attention in flood mapping. For this reason it could be useful to add a sentence on the use of Facebook and Flickr (with references) in this context since only Twitter based studies are addressed."*

15  In two of the studies that are included in this paragraph, Flickr data was used. We found no studies that used data from Facebook. This is likely a result of Facebook's restrictive policies, due to which only a very limited amount of data can be accessed through its API. On Facebook, a user of the API can only retrieve data about himself or his friends, whereas the APIs of Twitter and Flickr allow users to retrieve all public data from the platform. To clarify this, we made the following changes to the article:

20  - We added information about the data sources used in each of the studies mentioned
   - We appended the following to the paragraph: *"To our knowledge, no flood-related studies have used data from Facebook until now, which is likely due to Facebook being a more closed network. Flickr and Twitter allow for all public data to be found and extracted using their 'Application Programming Interfaces' (APIs; Interfaces to extract data from online platforms). The Facebook API however, is much more restrictive and cannot be used to retrieve*
25  *large amounts of public data."*

*"page 2; line 10-11: please rephrase."*

"User generated content" is a general term, often used in studies related to data from social media. In the remainder of the article, we have used the more specific term "social media content", since it was unclear to some readers who was meant by

the "user" in "user generated content". In order to make the article clear to both readers who are familiar with 'user generated content' as well as more novice readers, both terms are included. The first two sentences of the paragraph are used to link both. In order to clarify the first sentence, we made the following change:

- We replaced   by *"Data created by users of online platforms such as blogs, wikis and social media, often referred to as 'user generated content', offers an additional source of information about natural disasters."*

*"page 2; line11: "focus" at the past participle can take either double or single s, with the single option being highly preferred. Please consider changing it throughout the text."*

We changed all three mentions of "focussed" to "focused".

*"page 2; line14: The authors could be more precise when citing that "these data can be used to assess the extent of the disasters"? Can you provide some citation? In addition it would be useful to differentiate extent of the area, extent of damage and extent of human losses."*

This sentence is meant to build up to subsequent lines in which references are provided. By extent, we specifically mean the geographic extent of a disaster. To clarify this, we have changed the following:

- We replaced  by *"On a more detailed level these data have also been used to assess the geographic extent of a disaster."*

*"page 2, line 15: I would delete the world "merely" since it seems give a negative connotation to the sentence."*

We have deleted the word .

*"page 2; line 25: relay or rely? Please revise it carefully"*

We have made the following change to clarify that the flood maps created until now, did not include information about uncertainty:

- We changed  to *"...did not contain information about uncertainty..."*

*"page 3; line 17: can you be more precise on the amount of rainfall?"*

We have added information from the Met Office, about the amount of rainfall that fell over the two days prior to the flood, to the article. This involved the following change:

- We replaced " " by *"...caused by large amounts of rainfall, led to the flooding of a considerable area within the City of York in the North of England. Up to 120 mm of rain fell in Yorkshire over a 48 hr period between the 25th and 27th of December (Met Office, 2016)"*

*"page 3; line 17: over the month "of" December. Please consider adding the preposition "of"."*

This part of the sentence was removed in light of the previous comment.

*"page 3; line 18-19: "The 2015 floods" mentioned by Scott (2016) were registered all over UK or within the study area? Please consider rearranging this part by adding more information of the flood event considered (27th of December), adding some information about the damages and the area involved."*

Around the 27[th] of December, flooding occurred at multiple places in the North of England. The specifics about damages, other than that 453 homes and 174 businesses were flooded (Pidd, 2016) , are yet to be published. One year after the floods, the report by the York City Council is still being written (Pidd, 2016). To clarify this, we made the following change:

- We replaced ** by *"These large rainfall amounts resulted in the flooding of York and other places in the north of England. Within York, 453 residences as well as 174 businesses were flooded (Pidd, 2016). Detailed information about damages within York is yet to be published, since one year after the floods a report by the York City Council is still being written."* This made the reference to Stott (2016) obsolete.

*"page 3; line 25: "North of these ridges..", please rephrase this sentence"*

To improve readability we have changed the order of the words in this sentence:

- From ** to *"The inner-city of York is located to the north of these ridges."*

*"According to the methodology some issues need to be fixed, but first of all a general rearranging of the information in the sub-chapters needs to be done. Some details of the data are not concentrated together but are distributed on several chapters that make the reading a difficult process."*

To provide readers with an overview of all datasets used in the research, and the processing applied to the elevation data, we added the following table to the article:

| Data | Source | Purpose |
| --- | --- | --- |
| 2 m LIDAR DTM | EA (2014) | To group observations (Sect. 3.3) |
| | | To calculate water levels (Sect 3.3) |
| | | To estimate flood depth & extent (Sect 3.3) |
| | | To pinpoint Tweets referring to streets (Sect 3.2) |
| Twitter | Twitter streaming API | To extract flood observations (Sect. 3.2) |
| Google Maps | Used online | To find locations with Tweets (Sect. 3.2) |

| Google StreetView | Used online | To find exact locations of photographs (Sect. 3.2) |
| OpenStreetMap | Exported from osm.org | To simulate locational errors along streets (Sect 3.4) |
| Recorded historic flood outlines | EA (2015) | To evaluate flood extent in areas affected by non-fluvial flooding (Sect 3.5) |
| Recorded 2015 fluvial flood outline York (draft) | EA (Personal communication) | To evaluate flood extent in areas affected by fluvial flooding (Sect. 3.5) |

Along with this, we added a separate paragraph about the data to the beginning of section 3, and removing some of the then duplicate information in the subsequent paragraphs.

*"In addition, it is not clear how many tweets have been considered and/or point of interest have been derived."*

Since this is a result of the methods that are discussed in section 3.1 (Twitter data extraction), the number of Tweets that were found are reported in the results chapter (chapter four). However, the exact numbers of Tweets with each type of locational reference (POI/Street) is not reported. Therefore we changed the following at the beginning of the fourth chapter

- We replaced *"Although 56 of these Tweets had photographs attached, we could only match 26 of them to a location on Google StreetView."* by *"17 Tweets mentioned a point location (an address, intersection or POI) and 70 Tweets mentioned a street name, for which the elevation data was used to derive a point location. Although 56 Tweets from which locations were derived, had photographs attached, we could only match 26 of them to a location on Google StreetView."*

*"Secondly, it misses to mention the vertical accuracy of the DTM"*

We added information about the vertical accuracy of the DTM, originating from a report by the EA, in the new section about the data at the beginning of chapter 3 (Also see comment two comments above)

*"I think that the resampling method from 2 m to 20 m resolution should be stated before (please explain why)."*

We have included this information in the new section about the data at the beginning of chapter three. (Also see comment three comments above).

*"Moreover, when the observations have been grouped based on the LDDs have been based on the DEM or have been calculated by flow directions or by POI (point of interest) connected with flow directions?"*

The grouping was indeed done by combining the locations derived from the Tweets with the LDDs. We have clarified this by changing the following:

- We replaced *"This was done by calculating the LDDs in the area using the DTM. These LDDs were used to determine which cells are downstream of an observation."* by *"We grouped observations by combining information*

*about the LDDs in the area, which were derived from the DTM, with the locations of observations. The LDDs were used to determine which cells are downstream of an observation."*

*"In addition the authors need to justify the use of the IDW technique to determine the flood extent at this stage."*

We decided to use the IDW technique, instead of for example, the bilinear spline used by Fohringer et al. (2015), since we can apply smoothing using this technique. There is uncertainty in the water levels that are derived from Tweets, and by applying smoothing, the water levels of different observations that are in close vicinity, can be averaged. To clarify this, and put it in context with other studies, we have replaced the last paragraph of section 3.2 by:

- *"Previous studies applied both IDW (Werner, 2001) and bilinear spline interpolation (Fohringer et al., 2015) to calculate flood extents from irregularly spaced flood observations. We used IDW interpolation since it allows for smoothing, which is useful in averaging the water levels of clusters of uncertain flood observations from social media content. In case of certain flood observations, which should be followed exactly by the interpolated water surface, bilinear spline interpolation may be more appropriate. An additional advantage of IDW interpolation is that the nominator and denominator of Eq. (1) can be updated with new observations, meaning the additional computational time in real-time applications is limited. We slightly modified the method proposed by Werner (2001) to improve the realism of the interpolated water surface. Firstly, water levels were expressed relative to the elevation of the nearest drain instead of mean sea level. Secondly, observations were interpolated along their downstream flow paths and subsequently projected to the grid cells upstream of these flow paths, to create a grid of water levels. From this grid we subtracted the HAND map, to create an initial grid of water depths in the area. Since the water surface might be extrapolated to areas which were separated from the observations by small barriers, flooded areas that were not connected to any of the observations were removed, similar to the method suggested by Werner (2001). This procedure produced the deterministic flood maps."*

*"In addition try to explain the HAND map with few sentences for non-expert users. Some minor changes are listed as follows."*

To clarify what the HAND map is, we have added the following sentence after *"… inundation extents for fluvial floods.":*

- *"In contrast to a DTM, which contains elevation values relative to one single reference level, such as mean sea level, elevation values in a HAND map are relative to the nearest drainage channel. This drainage-normalized representation of the topography has a clear advantage for riverine flood extent mapping, as water depths over land can easily be related to water levels in the river."*

*"page 3; line 29: Which "useful information" have been extracted to create a deterministic flood extent estimate?"*

Since none of the Tweets about the floods in York gave an estimate of water depth, we only derived information about which locations were flooded from the Tweets. To clarify this, we have made the following change:

- We have replaced  by *"First we extracted locations where flooding was observed from flood-related Tweets".*

*"page 4; line 4: It is correct "to perform a step"? Please consider revising this sentence"*

We have changed the following:

- We have replaced  by *"The process used to create a database of Twitter based flood observations consisted of several steps"*

To clarify this sentence, we have made the following change:

- We have replaced  by *"We assumed that the water levels of flooded areas that are separated, are independent of each other. Therefore, we grouped observations to identify to which flooded area each observation belonged. The water levels of each group of observations were then interpolated separately"*

*"The chapter 3.4 "Evaluation of results" needs to be better contextualized and explained. The information inside this part seems to include limitation, problems, methodologies and expected results. I suggest rearranging this chapter."*

At the moment this paragraph contains both a discussion of the data, the methods and guidance on how to interpret the results. We have changed the following:

- We have moved the information about the validation data to a separate 'Data' paragraph at the beginning of the chapter: *"Recorded flood extents were used to validate the flood maps (Sect 3.5). A draft version of the fluvial flood extents of the City of York was supplied by the EA. These flood extents only identified areas that were directly affected by flooding from the rivers. However, areas separated from the river around Knavemire Road, Water Lane and Shipton Road were also known to be flooded based on news articles. The flood extents around these locations were approximated by using the EA dataset of recorded flood extents (EA, 2015) from between the years 1991 – 2012. These were merged with the recorded fluvial flood extent from 2016 into one validation dataset."*
- We have removed the sentences about the interpretation of the reliability diagram, since this becomes clear in the results chapter.

*"In addition try to better explain the F(2) statistic expressed at page 7; line 20."*

A long textual explanation of the F(2) statistic will impact the readability of the article. Therefore we have removed the sentence that textually explains the F(2) statistic, and instead introduce the equation for the F(2) statistic, with a short explanation of the terms in this equation (for the revised paragraph: see 'track-changes' supplement).

*"The discussion, should provide some interpretation of the results emerged without wasting entire sentences restating the results (ex, page 12; line 11-12 among others). Please, try to verify it and do not repeat same concepts."*

Some concepts are indeed repeated in the current discussion section, making the section unnecessarily lengthy and reducing the readability. To improve the readability of the discussion and remove the duplications from the discussion paragraph, we have completely rewritten the discussion chapter (5).

*"In addition, you might relate your work to the findings of other studies by finding crucial information in someone else's study that helps you interpret your own data, or perhaps you*

*will be able to reinterpret others' findings in light of yours. In either case you should discuss reasons for similarities and differences between yours and others' findings."*

We found only very little studies regarding probabilistic flood inundation maps that used a validation method similar to ours.

We have changed the following:

- We added to the discussion chapter: "*A comparison to the work of Giustarini et al. (2016), who produced probabilistic flood maps from synthetic aperture radar (SAR) data and used the same validation technique, indicates that results are similar. It illustrates that probabilistic flood maps from SAR data provide a degree of accuracy comparable to the ones in our study, with probability-error values up 0.38. Although their reliability diagrams differed among case studies, none of them had a consistent overestimation of flood probability in all bins of the reliability diagram, like the ones from social media content. This indicates that the method presented in this paper still has some limitations.*"

*"In addition, please rephrase page 12; lines 2-4"*

10   In light of a previous comment (two comments above), the discussion was rewritten.

*"Please, cite in the text the authors of the script just mentioned with a doi at page 13; line 15."*

We created the scripts we used for the analyses in this paper ourselves. To clarify this we have made the following change:

- Add a reference to the text (and in the list of references): replace "" by "*...code used for the different analyses (Brouwer, 2016) is...*"

*"Figure 1 needs more details of where the study area is located. A smaller map of UK where the study area is highlighted is highly suggested to help readers in locating the area. Terrain elevation needs "m (a. s. l.)"."*

This indeed makes the location of the study area more clear, for readers who are not familiar with the UK. We have replaced

15   Figure 1 with the following figure:

[Figure]

Since this is not strictly speaking a frequency (like $s^{-1}$ for example), but a number of observations, we propose to replace Figure 5 with the following figure:

[Figure]
* * *
*"Figure 6. What's the confusion matrix? Please add in the caption what represent the numbers in square brackets."*

5    It is indeed unclear what the confusion matrix is. The concept 'confusion matrix' is however not important for understanding figure 6, since the meaning of the colours in the map can be derived from the legend. Also, it is not completely clear that the numbers in brackets are used to indicate locations in the map. Therefore, we have made the following changes:

- We removed in section 4.2: ""
10
- We changed the caption of Figure 6 to: "*Comparison between the deterministic flood map (modelled) and validation data (observed). The locations denoted by the numbers [1] to [4] are referred to in text.*"

*"Figure 7. Please add in the caption what represent the numbers in square brackets."*

We have appended the following to the figure caption:

- *"The locations denoted by the numbers [1] to [3] are referred to in text."*

*"Figure 8. I suggest enlarging this figure to be able to appreciate the details."*

We have replaced figure 8 with the following figure:

[Figure]

**Response to RC2**

*"What is the F(2) statistic is explained in the text (section 3.4), but not in the abstract. Please add a definition when mentioning F(2) in the abstract."*

A full explanation of the F(2) statistic within the abstract is not feasible, since it would involve multiple lines of text. This full description is included in chapter 3 of the paper. To indicate the relevance of the $F^{(2)}$ value in the abstract, we have made

5 the following change:

- We have replaced *"...($F^{(2)}$ = 0.69)..."* by *"...($F^{(2)}$ = 0.69; a statistic ranging from 0-1, with 1 expressing a perfect fit with validation data)..."*

*"Explain why the DWD has been estimated to range between 20 cm and 80 cm (section 3.3)"*

This is indeed unclear. The decision to use this range was mainly based on photographs that were found in news articles of the floods. Therefore, we have replaced lines 6&7 on page 7 with:

10 - *"Based on photographs in news articles about the flooding in York, the water depth in most places was estimated to be between 20 and 80 cm. Therefore, the DWD was varied between 20 and 80 cm. For the smoothing and power parameter, no clear information about the range was available. Errors in these parameters were simulated using the rather conservative ranges of 0-2000m and 2-5 respectively. A uniform distribution was used to simulate errors in the DWD, range- and power- parameters, since there was no specific information available regarding their error*
15 *distributions."*

*"I understand that the (potentially large) uncertainties of the validation data were ignored when building the reliability diagrams. I therefore feel that the uncertainties of the probabilistic maps may be even more overestimated than claimed in the paper. Please*

> *discuss this issue, the possible consequences of neglecting the validation data uncertainties, and how they could be included in the analysis."*

We did not mention the uncertainties of the validation data in the article, although these might be important in evaluating probabilistic flood inundation maps. Therefore, we have made the following change to the article:

- We have replaced lines 13-16 on page 11 with: *"Furthermore, the results of the analysis could have been affected by the quality of the maps used for validation. The data for validating the river flood extents was created from a combination of ground observations and aerial photography. In places flooded separately from the river however, recorded historic flood extents were used, which might have been in accurate. However, actual observed flood extents for 2015 were used for the majority of the area. Therefore, we have no reason to believe that there are large uncertainties in the validation data."*

**Other changes**

In addition to the changes that were made in response to the referee comments, we have also implemented the following changes:

- We changed a substantial part of the abstract, to improve readability.
- In addition to the change made in response to a comment of anonymous referee 1, a part of the paragraph about the grouping of observations (section 3.3, formerly 3.2) was rewritten.
- In rewriting the discussion paragraph, in response to some of the comments in RC1, we have made the following sections
    1. *Potential*: Discussing the potential of the method
    2. *Limitations*: Discussing the current limitations of the method
    3. *Recommendations*: Discussing the recommendations for future research
- In light of the restructuring of paragraph 3, following comments of anonymous referee 1, figure 2 was slightly modified, and does not include the filtering based on dates as a separate step now:

[Figure]

- All mentions of 'manuscript' were changed to 'paper'
- We have implemented some minor text changes, to improve readability.

[revised manuscript text omitted]

```
┌─────────────────────────────────┐
│      All English Tweets          │
│      (25 to 30-12-2015)          │
└─────────────────────────────────┘
              │
              ▼
┌─────────────────────────────────┐
│      Select flood related Tweets │
└─────────────────────────────────┘
              │
              ▼
┌─────────────────────────────────┐
│   Select Tweets referring to York│
└─────────────────────────────────┘
              │
              ▼
┌─────────────────────────────────┐
│ Select Tweets with detailed locations│
└─────────────────────────────────┘
        │                  │
        ▼                  ▼
┌──────────────┐   ┌──────────────────┐
│ Derive locations│ │ Derive photo     │
│ from text      │  │ locations (validation)│
└──────────────┘   └──────────────────┘
        │                  │
        └────────┬─────────┘
                 ▼
┌─────────────────────────────────┐
│         Twitter dataset          │
└─────────────────────────────────┘
```

5    **Figure 2.** Process of constructing the dataset of Tweets

[Figure]

*"**Cumberland Street** in York - say they're used to flooding here but only 2000 was worse"*

Attached Photograph          Google StreetView

**Figure 3.** Example of determining the error in the spatial coordinates derived from the text of a Tweet, based on an attached photograph. The grey dot is the location derived from the text of the Tweet, and the black dot is the location derived from the attached photograph.

[Figure]

5    **Figure 4.** Process of creating flood extent maps

[Figure]

**Figure 5.** Locational errors in X/Y coordinates of all Tweets (a), only the ones referring to streets (b) and only the ones referring to point locations (c)

[Figure]

**Figure 6.** A comparison between the deterministic flood map (modelled) and validation data (observed). The locations denoted by the numbers [1] to [4] are referred to in text.

[Figure]

**Figure 7.** Probabilistic flood map generated by simulating locational errors, errors in elevation data and errors in parameters (a) and a probabilistic map generated simulating errors in the elevation data and parameters only (b). The locations denoted by the numbers [1] to [3] are referred to in text.

**Figure 8.** Reliability diagrams constructed from the probabilistic flood map generated by simulating all errors (a) and by simulating only errors in the elevation data and parameters (b). The small histograms give the number of cells within each 10% bin of modelled flood probability.

[Figure]

**Figure 9.** Empirical cumulative distribution functions of the $F^{(2)}$ performance statistics derived by simulating purely errors in the elevation data (blue), errors in the parameters (red), locational errors (black) and the combination of these errors (dashed green).

**Table 1.** Datasets used in this study

| Data | Source | Purpose |
| --- | --- | --- |
| 2 m LIDAR DTM | EA (2014) | To group observations (Sect. 3.3) |
| | | To calculate water levels (Sect 3.3) |
| | | To estimate flood depth & extent (Sect 3.3) |
| | | To pinpoint Tweets referring to streets (Sect 3.2) |
| Twitter | Twitter streaming API | To extract flood observations (Sect. 3.2) |
| Google Maps | Used online | To find locations with Tweets (Sect. 3.2) |
| Google StreetView | Used online | To find exact locations of photographs (Sect. 3.2) |
| OpenStreetMap | Exported from osm.org | To simulate locational errors along streets (Sect 3.4) |
| Recorded historic flood outlines | EA (2015) | To evaluate flood extent in areas affected by non-fluvial flooding (Sect 3.5) |
| Recorded 2015 fluvial flood outline York (draft) | EA (Personal communication) | To evaluate flood extent in areas affected by fluvial flooding (Sect. 3.5) |

**Table 12.** Quantification of error sources. [a]See Sect. 4.23

| Error source | Distribution | Parameter | Value |
|---|---|---|---|
| Elevation data | Normal (spatially auto correlated) | μ: | 0 m |
| | | σ: | 0.2 m |
| | | Corr. distance: | 100 m |
| Tweets (point location)[a] | Normal (X/Y coordinate) | μ: | 0 m |
| | | σ: | 50 m |
| Tweets (Street location)[a] | Normal (along street) | μ: | 0 m |
| | | σ: | 200 m |
| Power parameter | Uniform (integers only) | Lower bound: | 2 |
| | | Upper bound: | 5 |
| Smoothing parameter | Uniform | Lower bound: | 0 m |
| | | Upper bound: | 2000 m |
| DWD | Uniform | Lower bound: | 0.2 m |
| | | Upper bound: | 0.8 m |

---

## Author Response (AR2)

**Response to referee comments on "Probabilistic Flood Extent Estimates from Social Media Flood Observations" by Tom Brouwer et al."**

5 The authors would like to thank the referees for their reviews of the revised manuscript. Anonymous referee # 1 suggested some minor changes to be made to the article before publication. Our response to these comments is given in the paragraph below. This paragraph is followed by a track-changes version of the manuscript, to detail the changes in-text.

**Response to anonymous referee # 1**

*"pg 24; line 27-28: please rephrase"*

10 Since the word "however" was repeated in two consecutive sentences, these sentences have been rewritten:

- "*Even though the use of historic data to validate flood extents in places that were flooded separately from the river might have been inaccurate, actual observed flood extents for 2015 were used for the majority of the area*"

*"pg 26; line 2:.."that" can be produced. Please consider to add "that""*

15 To further clarify the sentence, the word "that" is inserted, but moved more to the front of the sentence:

- "For real-time applications, it is vital to collect a high number of observations to ensure that accurate and up to date maps can be produced at any point in time."

*"pg 26; line 3: "selected" could sound better that "selection"*

This sentence is specifically about the search technique used to find relevant Tweets. This is referred to as
20 "selection techniques" in this sentence, although "search technique" is more clear. Therefore this sentence has been changed to:

- "The search technique used in this paper was only able to find a small number of tweets"

*"Table 1: Was Google maps used to find locations "with" twitter or "of" twitter?"*

The word "of" is better, this has been changed.

*"Table 2: For ease of reading I suggest to merge columns "parameters" and "value" together and try to enrich the caption a bit more."*

The last two columns have been merged. This caption now better describes the table:

- "Error distributions used to simulate sources of error"

**Other changes**

To further clarify figure 1, a small map detailing the location of the study area within the UK has been added.

**Track-Changes version of the manuscript**

The track-changes version of the manuscript can be found starting from the next page.

[revised manuscript text omitted]

| Error source | Distribution | Parameter values |
| --- | --- | --- |
| Elevation data | Normal (spatially auto correlated) | μ: 0 m
σ: 0.2 m
Corr. distance: 100 m |
| Tweets (point location)[a] | Normal (X/Y coordinate) | μ: 0 m
σ: 50 m |
| Tweets (Street location)[a] | Normal (along street) | μ: 0 m
σ: 200 m |
| Power parameter | Uniform (integers only) | Lower bound: 2
Upper bound: 5 |
| Smoothing parameter | Uniform | Lower bound: 0 m
Upper bound: 2000 m |
| DWD | Uniform | Lower bound: 0.2 m
Upper bound: 0.8 m |